# Application of UPLC-QTOF-MS Based Untargeted Metabolomics in Identification of Metabolites Induced in Pathogen-Infected Rice

**DOI:** 10.3390/plants10020213

**Published:** 2021-01-22

**Authors:** Mira Oh, SeonJu Park, Hun Kim, Gyung Ja Choi, Seung Hyun Kim

**Affiliations:** 1College of Pharmacy, Yonsei Institute of Pharmaceutical Sciences, Yonsei University, Incheon 21983, Korea; purunmei2002@naver.com; 2Chuncheon Center, Korea Basic Science Institute (KBSI), Chuncheon 24341, Korea; sjp19@kbsi.re.kr; 3Center for Eco-Friendly New Materials, Korea Research Institute of Chemical Technology, Daejeon 34114, Korea; hunkim@krict.re.kr (H.K.); kjchoi@krict.re.kr (G.J.C.)

**Keywords:** LC-MS, plant-pathogen interaction, metabolic change, pathway analysis, principal component analysis (PCA)

## Abstract

Metabolomics is a useful tool for comparing metabolite changes in plants. Because of its high sensitivity, metabolomics combined with high-resolution mass spectrometry (HR-MS) is the most widely accepted metabolomics tools. In this study, we compared the metabolites of pathogen-infected rice (*Oryza sativa*) with control rice using an untargeted metabolomics approach. We profiled the mass features of two rice groups using a liquid chromatography quadrupole time-of-flight mass spectrometry (QTOF-MS) system. Twelve of the most differentially induced metabolites in infected rice were selected through multivariate data analysis and identified through a mass spectral database search. The role of these compounds in metabolic pathways was finally investigated using pathway analysis. Our study showed that the most frequently induced secondary metabolites are prostanoids, a subclass of eicosanoids, which are associated with plant defense metabolism against pathogen infection. Herein, we propose a new untargeted metabolomics approach for understanding plant defense system at the metabolic level.

## 1. Introduction

Plants constantly interact with diverse microorganisms during their lifetime. Within the plant environment, not only nonpathogenic microbes but also pathogenic microbes can establish intricate interactions with their hosts. Plants recognize invading phytopathogens and resist their attack by inducing rapid defense responses [1]. Induction of metabolites, which act as plant defense compounds, is the most common and important part of the plant defense repertoire [2]. Plant–pathogen relationships are worth studying in terms of biological importance and metabolite richness [3]. Studying the types of metabolites produced in plants affected by phytopathogens provides understanding of the plant defense mechanisms and facilitates the discovery of new defense metabolites.

Metabolomics technology offers a meaningful approach for the comprehensive profiling and comparison of metabolites in biological systems. Metabolomics offers the unbiased ability to differentiate genotypes based on the metabolite level, even if there are no visible phenotype differences. As a diagnostic tool, metabolomics coupled with various spectroscopic analyses help to determine the mode of action of various herbicides or drugs on plants or humans, respectively [4,5,6]. There are two principal metabolomics approaches: targeted and untargeted. The latter approach aims to gather as much information on metabolites as possible by taking all information present in datasets into account, thus it is the most rational way to identify a wide variety of metabolite changes in plants [7]. Several plant metabolomics studies have been conducted with an untargeted approach [8,9,10]. To date, various analytical techniques have been applied to metabolomics. Although nuclear magnetic resonance is the most uniform detection technique, it has a lower sensitivity compared to mass spectrometry (MS)-coupled techniques, and the detection ability of low-abundance metabolites is limited. For these reasons, in recent times, high-resolution mass spectrometry (HR-MS) and tandem mass spectrometry (MS/MS) are widely used in metabolomics [11]. HR-MS, such as quadrupole time-of-flight mass spectrometry (QTOF-MS), provides accurate mass and specific fragment patterns of MS/MS, which can improve the speed and the efficiency of metabolite identification.

Rice blast, caused by the rice blast fungus (*Magnaporthe oryzae*, Magnaporthaceae), is extremely difficult to control. It causes significant economic and humanitarian problems worldwide [12]. To control rice blast, most of the studies have focused on understanding the defense mechanism of rice at transcriptional, proteomic, or biophysical levels [13,14,15,16]. Recently, as metabolomics has become a widely used tool for high-throughput research in the field of plant science, metabolomics studies on rice–pathogen interaction have also been undertaken [17,18]. Metabolism is the final step of the biochemical dynamics of living organisms. Thus, understanding how metabolites change is essential in studying plant defense mechanisms in system biology [2,19]. Plant metabolomics is growing as an essential tool for studying system biology in plant science, especially for crop enhancement [7]. However, there are only a few metabolomics studies investigating the actual metabolite changes caused by pathogen infection, which is the last step in the biochemical pathway of plant defense mechanism. Therefore, comparing the metabolites of pathogen-infected plants with control could be an ideal approach to elucidate the biochemical pathways involved in plants’ multi-factor defense mechanisms. Herein, we applied liquid chromatography (LC)-QTOF-MS based metabolomics analysis for mass feature profiling in control and pathogen-infected rice. The most differentially induced metabolites in infected plants were identified using computational annotation tools, and pathway analysis was performed. This approach may provide a new perspective for understanding plant defense mechanisms and facilitating the discovery of unknown defense compounds.

## 2. Results and Discussion

### 2.1. LC-MS Based Untargeted Metabolomics

As part of their defense mechanisms, plants respond to invading pathogens by producing diverse compounds. The primary metabolites induced as a by-product of plants’ defense metabolism have been frequently studied [20,21]. Meanwhile, secondary metabolites involved in plants’ defense metabolism have been relatively understudied due to their structural complexity and diversity and the consequent difficulty of identification. However, recently developed metabolomics approaches using spectrometric techniques have enabled easy and rapid analysis of secondary metabolites [22].

LC-MS total ion current (TIC) chromatograms were acquired from the collected control and pathogen-infected rice samples for spectrometric analysis of metabolites. Metabolites were more strongly ionized in the ESI positive mode (Figure 1A,B) than the negative mode (Appendix A). Thus, raw LC-MS data in the positive mode were applied for further metabolomic analysis. In addition, three-dimensional plots of raw LC-MS data in the positive ionization mode were obtained using Mzmine 2.53 3D visualizer, which showed peak patterns of samples more clearly (Figure 1C,D).

### 2.2. Mass Differences between Two Rice Groups

Multivariate statistical analysis was used to confirm significant differences in mass values between the two groups (control and infected rice). The mass peak list of twelve samples (six samples from each group) containing 328 mass features was uploaded for principal component analysis (PCA) and partial least squares discriminant analysis (PLS-DA). PCA is an unsupervised technique used for dimensionality reduction of multivariate data while preserving most of the variance. PLS-DA is a supervised chemometric method used to optimize separation between different groups [23]. The PCA score plot (Figure 2A) showed two clusters of samples indicating the differentiation between the two groups. However, 95% of the cluster’s confidence regions were overlapped, indicating that, although PCA cannot clearly explain the differences between the control and the infected rice, the means may be significantly different from one another at the α = 0.05 level [24]. The two groups were more completely clustered in the PLS-DA than the PCA (Figure 2B), showing clear separation with 95% confidence regions. The cross-validated coefficient of determination for the model with five components was Q2 = 0.955. Despite the TIC chromatograms’ similarity between two groups (Figure 1A,B), multivariate analysis results showed statistical differences of mass profiles between control and infected rice (Figure 2A,B). Cluster separation results were also confirmed by a heatmap made using Ward’s hierarchical clustering algorithm (Figure 2C). Since the current study aimed to identify the most differentially induced metabolites in infected rice compared to a control, we obtained a list of the most induced mass through a volcano plot analysis based on both fold change (FC) analysis and t-tests (Figure 2D). It showed 61 mass features that were induced in infected rice. Twelve major discriminant features were selected from the list based on FC (>2) and *p*-value (<0.1) thresholds. Selected metabolites were arranged sequentially according to their log2(FC) values (Table 1), and the relative peak height of the metabolites was represented using box and whisker plots (Appendix A).

### 2.3. Putative Identification of Differentially Induced Metabolites

Metabolites were identified based on various spectra databases. The exact mass and the chemical formula were calculated by MassHunter software, and several compounds with similar masses and formulas were selected as candidates. The metabolites were then finally annotated by comparing the MS/MS fragmentation patterns. The most highly induced metabolite with the value of log2(FC) 20.503 among the metabolites induced in infected rice was *m*/*z* 317.2096 and tentatively identified to be 15-deoxy-Δ^12,14^-prostaglandin J2 (PGJ2) through MS/MS fragment comparison (Figure 3). In the same way, with the exception of *m*/*z* 383.2025, 303.1402, and 699.3502, the other eight induced metabolites were tentatively identified by comparing their MS/MS patterns (Table 1). The chemical structures of the identified metabolites are shown in Figure 4.

In order to characterize the class of unidentified compounds of *m*/*z* 383.2025, 303.1402, and 699.3502, molecular networking was performed. The molecular network is a spectral analysis tool used to visually display the chemical space present in the tandem mass spectra of small molecules, which enables the mapping of the chemical diversity observed in untargeted mass spectrometry experiments. The global natural products social molecular networking (GNPS) is a public infrastructure that enables molecular networking [25]. GNPS aligns each MS/MS spectrum in a dataset to each of the others and assigns a cosine score to each combination to describe their similarity. Structurally related molecules are clustered using the MS-Cluster algorithm. Classic molecular networking was generated using the raw LC-MS data of control and infected rice and was visualized as a network of nodes and edges with Cytoscape 3.8.0, an open-source software for visualizing complex networks [25] (Figure 5A). Of 594 nodes, 272 nodes were shared between two groups, and 50 more nodes were observed in the infected rice than in control (Figure 5C). Unfortunately, mass features of *m*/*z* 383.2025, 303.1402, and 699.3502 did not belong to specific classes in the classic molecular network. Recently, network annotation propagation (NAP) has been introduced as an alternative annotation method, which uses in silico prediction with a re-ranking system to increase annotation accuracy [26]. Although NAP cannot provide the level of confidence required for the exact identification of compounds, it enables a class level of annotation and candidate prioritization of unknown fragmented mass spectrum [27]. Using NAP, the consensus candidates of *m*/*z* 383.2025 were characterized to be oxadiazoles, but *m*/*z* 303.1402 and 699.3502 were not categorized in silico (Figure 5B). The molecular networking job on GNPS and NAP can be found at https://gnps.ucsd.edu/ProteoSAFe/status.jsp?task=b7d63e4ac9164971baf712b9752061c3 and https://proteomics2.ucsd.edu/ProteoSAFe/status.jsp?task=d5707bfad1a94efbb6108759e47a0d4a, respectively.

### 2.4. Metabolic Pathway Analysis

To determine the plant metabolic pathway activated in infected rice, pathway analysis was conducted. Of the twelve most differentially induced metabolites, nine were annotated with KEGG, HMDB, or PubChem ID (Appendix A) and then submitted to MetaboAnalyst pathway analysis based on the KEGG plant database. The following ten KEGG plant pathways were detected: phenylalanine metabolism; phenylalanine, tyrosine, and tryptophan biosynthesis; aminoacyl-tRNA biosynthesis; linoleic acid metabolism; tropane, piperidine, and pyridine alkaloid biosynthesis; zeatin biosynthesis; tryptophan metabolism; glycine, serine, and threonine metabolism; phenylpropanoid biosynthesis; purine metabolism (Figure 6, Table 2). The pathways were related to five pathway modules and one genetic information processing pathway (Table 3).

Among nine identified compounds, amino acids phenylalanine, tryptophan, and the phenylalanine derivative, phenylacetaldehyde, were found to be related to amino acid metabolism consisting of phenylalanine metabolism, phenylalanine, tyrosine, and tryptophan biosynthesis, tryptophan metabolism, and glycine, serine, and threonine metabolism. Phenylalanine was also found to be involved in the biosynthesis pathway of secondary metabolites, especially tropane, piperidine, and pyridine alkaloid biosynthesis and phenylpropanoid biosynthesis. In plants, phenylalanine and tryptophan are pivotal precursors for secondary metabolite formation, such as indole alkaloids, phenylpropanoids, flavonoids, and the phenolic polymer lignin [28,29]. Secondary metabolites biosynthesized from tryptophan via two major intermediates, indole-3-acetaldoxime and tryptamine, are scattered throughout the plant kingdom, and some of them are implicated in plant defense mechanisms based on their antimicrobial activity [29,30]. Ueno et al. [31] pretreated rice leave with indole-3-acetic acid (IAA), tryptamine, or tryptophan before infecting them with the rice blast fungus *M. oryzae*, and the blast lesion formation was suppressed in IAA and tryptophan treated leaves compared to control. From the study, the accumulation of tryptophan-derived secondary metabolites reduced the damage caused by a fungal infection in rice leaves, which suggested that tryptophan and tryptophan-derived secondary metabolites function as part of the effective defense mechanism of rice. In addition, phenylalanine-derived secondary metabolites accumulated via phenolic compound biosynthetic pathways significantly impact plants’ defensive response to pathogenic infection [28,32].

Zeatin, a member of the cytokinin family, is a phytohormone that is involved in various growth and development processes in plants [33]. Zeatin-type cytokinin regulates plant immunity against pathogens by suppressing symptom development and restraining the pathogen-induced cell death response in plants [33,34]. In the present study, we hypothesized the activation of amino acid-related metabolism and the zeatin biosynthesis pathway to be a result of the induced plant defense mechanism in infected plants. The purine metabolism and the aminoacyl-tRNA biosynthesis pathways were also activated in infected rice by producing adenine and amino acids such as phenylalanine and tryptophan, respectively.

Plant oxylipins, a large family of metabolites derived from polyunsaturated fatty acids, represent a vast and diverse family of secondary metabolites, which contribute to plants’ local and systemic defense mechanisms [35]. Some oxylipin profiling studies report that a few genes encoding oxylipin biosynthetic enzymes are specifically induced in pest or pathogen-inoculated plants, and the production of oxylipin was increased [36,37]. In accordance with previous studies, the results indicated that four among twelve most induced metabolites in infected rice belong to the oxylipin family, prostanoids, namely, PGJ2, prostaglandin F1α, 11-dehydro-thromboxane B3, octadecanoid, and 9-hydroperoxy-10*E*,12*Z*-octadecadienoic acid (9-HPODE). In particular, 9-HPODE has been demonstrated to produce a hypersensitive response cell death in pathogen-infected plants [38]. In the lipid metabolism process, 9-HPODE is biosynthesized from linoleic acid and catalyzed by lipoxygenase. Lipoxygenase induction is increased in response to wounding or herbivore attack [39]. Similarly, both prostaglandins and thromboxanes are derived from arachidonic acid by cyclooxygenase in the initial stage of arachidonic acid metabolism [40]. These results suggest that pathogen-infected rice activates lipid metabolism that is involved in the production of its by-products, prostanoids and octadecanoid, which are the most highly induced metabolites in pathogen-infected rice compared to that of control.

## 3. Materials and Methods

### 3.1. Host Plant and Fungal Pathogen

Disease development using the rice (*Oryza sativa*, Poaceae) cultivar, Chucheong, as the host plant was performed as previously described [41]. Briefly, two or three leaf stages of rice plants grown in a greenhouse at 25 ± 5 °C for three weeks were inoculated by spraying with a spore suspension of *Magnaporthe oryzae* KACC 46552 (Magnaporthaceae). The fungal strain was provided by the Korean Agricultural Culture Collection (Jeonju, Korea). Sporulation and maintenance were performed on oatmeal agar plates (5% oatmeal (*w*/*v*) and 2% agar (*w*/*v*)) [42]. For the inoculum, a concentration of spore suspension (5 × 10^5^ spores/mL) was adjusted in 0.025% aqueous Tween 20 solution. Inoculated plants were incubated in a humidified chamber (25 °C) for one day and then transferred to a growth chamber (25 °C and 80% relative humidity) for four days of incubation. As a control, the same stage of rice plants were sprayed with 0.05% Tween 20 solution and incubated as described above.

### 3.2. Sample Preparation

Rice leaves were collected under controlled conditions, flash-frozen, and stored at −80 °C. Leaves were ground to a fine powder with a mortar and pestle in liquid nitrogen. Leaf powder (10 mg) was transferred to a 5 mL glass vial, and 1 mL of methanol was added. All processes were carried out under liquid nitrogen until the addition of solvent to the sample vial. All experimental tools (such as glass vial and spatula) were kept frozen in liquid nitrogen throughout the experiment. The sample was vortexed for 30 s and sonicated for 30 min at room temperature. The supernatant was then filtered using a 0.2 μm pore syringe filter (Whatman, Clifton, USA). The filtrate was then completely evaporated under a nitrogen flow and stored at −20 °C until analysis. For LC-MS analysis, the dried sample was weighed and dissolved in LC-MS grade methanol (JT Baker, Phillipsburg, USA) to a concentration of 1 mg/mL.

### 3.3. UPLC-QTOF-MS Analysis

All samples were analyzed by an ultra-high performance liquid chromatography (UPLC)-QTOF-MS analytical system. The instrument consisted of an Agilent 1290 Infinity LC system (Agilent technologies, Palo Alto, CA, USA) coupled with an Agilent 6550 iFunnel QTOF LC/MS system equipped with dual Agilent Jet Stream (AJS) ESI source. Each sample was injected in six replicates at a volume of 10 μL. Blank (100% MeOH) was run at the beginning of the sample sequence for the elimination of background features. Metabolite separation was performed using a YMC-Triart C18 column (2.0 × 150 mm, 1.9 μm; YMC KOREA Co., Seongnam, Korea) at 25 °C. The mobile phases were 0.1% formic acid in water (A) and 0.1% formic acid in ACN (B) with the following gradients: 5–95% B (0–20 min), 95–100% B (20–20.1 min), 100% B (20.1–23 min), 100–5% B (23–23.1 min), 5% B (23.1–25 min). The flow rate was 0.4 mL/min.

The MS experiment was performed with a dual AJS ESI source under the following conditions: drying gas temperature 300 °C, drying gas flow 8 L/min, nebulizer gas pressure 35 psi, sheath gas temperature 350 °C, sheath gas flow 11 L/min, and capillary voltage +3.5 kV and −3.5 kV for the positive and the negative ionization modes, respectively.

QTOF parameters were set with a mass range of 100−1000 *m*/*z* and an acquisition rate of five spectra/sec for MS and a mass range of 40−1000 *m*/*z* and an acquisition rate of three spectra/sec for MS/MS. MS/MS fragment patterns were obtained using fixed collision energies of 20 eV and 40 eV. Mass calibration was performed with an Agilent tune mix (Agilent technologies, Palo Alto, CA, USA) from 100 to 1600 Da, and the data were acquired in centroid mode using a high-resolution mode (4 GHz).

### 3.4. Feature Finding

Mass feature detection was performed using MZmine 2.53 [24]. The mzML files were imported and cropped to a retention time range of 0–20 min. The mass detection noise level was 1500 for MS1 and 20 for MS2. Chromatograms were built with a minimum time span of 0.01 min, a minimum height of 5000, and an *m*/*z* tolerance of 0.001 *m*/*z* (or 5 ppm). Chromatogram deconvolution, which separates each detected chromatogram into individual peaks, was achieved using a baseline cut-off algorithm with a minimum peak height of 2500, a peak duration range of 0.02–0.4 min, and a baseline level of 500. Deconvoluted peaks were deisotoped using the isotopic peaks grouper algorithm with an *m*/*z* tolerance of 0.006 (or 10 ppm) and a retention time tolerance of 0.15 min. Peaks were aligned in a peak table using the join aligner module, which aligns detected peaks with a match score with an *m*/*z* tolerance of 0.006 (or 10 ppm), an absolute retention time tolerance of 0.3 min, an *m*/*z* weight of 70, and a retention time weight of 30. Contaminated features identified by blank injection and duplicated peaks were manually removed from the aligned peak table. Feature tables were then filtered to include only features that contained a minimum of three peaks in a row.

### 3.5. Statistical Data Analysis

MetaboAnalyst 4.0 (http://www.metaboanalyst.ca) was used for univariate, multivariate, and clustering analyses [43]. The peak intensity table exported from MZmine software was uploaded and normalized with a control group’s probabilistic quotient normalization. Volcano plot analysis was conducted based on fold change (FC) analysis and t-tests (*p* < 0.1). For the FC analysis, the data before column normalization were used to compare the absolute value changes between two group means. Principal component analysis (PCA) and partial least squares discriminant analysis (PLS-DA) were performed using R package’s prcomp and plsr functions, respectively. For heatmap clustering analysis, Euclidean’s distance measure method and Ward’s hierarchical clustering algorithm was used based on the hclust function of the stats R package.

### 3.6. Annotation of Metabolites and Metabolic Pathway Analysis

The most differentially induced metabolites of infected rice compared with control were selected based on multivariate analysis results, and their annotation was made according to exact mass, chemical formula, and MS/MS fragmentation spectra. Chemical formulas were calculated with a mass accuracy of <7.5 ppm using MassHunter qualitative analysis software (Version B.06.00, Agilent Technologies, Palo Alto, CA, USA). For metabolite annotation, our MS/MS fragmentation data were compared with those of Metlin (http://metlin.scripps.edu/), MassBank (http://massbank.jp/), HMDB (http://www.hmdb.ca/), and GNPS (https://gnps.ucsd.edu/) spectra databases. Pathway analysis of induced metabolites in infected rice was conducted using the pathway analysis module of MetaboAnalyst 4.0. The pathways were analyzed based on the KEGG pathway database (https://www.genome.jp/kegg/pathway.html), using the hypergeometric test for over-representation analysis and out-degree centrality for pathway topology analysis.

## 4. Conclusions

In this study, rice plants’ metabolite changes resulted from the infection of *M. oryzae* were investigated. Twelve metabolites were selected as the most induced metabolites in pathogen-infected rice compared to a control. Among them, nine of the metabolites were annotated using spectral databases. Pathway analysis revealed that most of the nine highly induced metabolites are associated with plant defense metabolism. In particular, the activation of lipid metabolism may explain the induction of prostanoids and octadecanoids, which were the most highly induced secondary metabolites in the infected rice. These results showed actual metabolite changes in pathogen-infected rice at the secondary metabolite level. The use of untargeted metabolomics in infected plants may provide a new perspective and further understanding of plant defense mechanisms.

## Figures and Tables

**Figure 1 plants-10-00213-f001:**
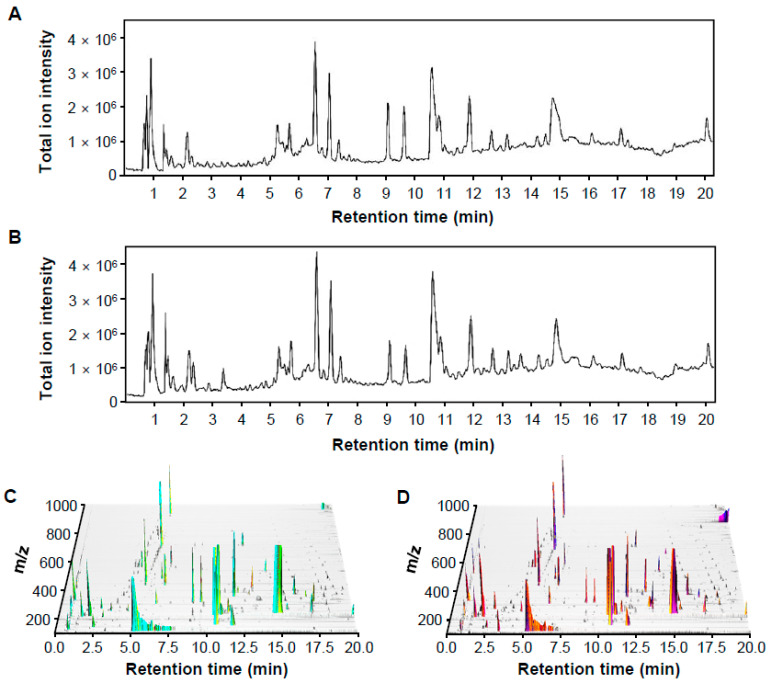
Representative LC-MS chromatographic profiling of control and pathogen-infected rice in positive ion mode. (**A**,**B**) Total ion current (TIC) chromatograms of control (**A**) and infected (**B**) rice. (**C**,**D**) 3D plot chromatograms of control (**C**) and infected (**D**) rice, visualized by MZmine 2.53 3D visualizer. Z axis represents the signal intensity.

**Figure 2 plants-10-00213-f002:**
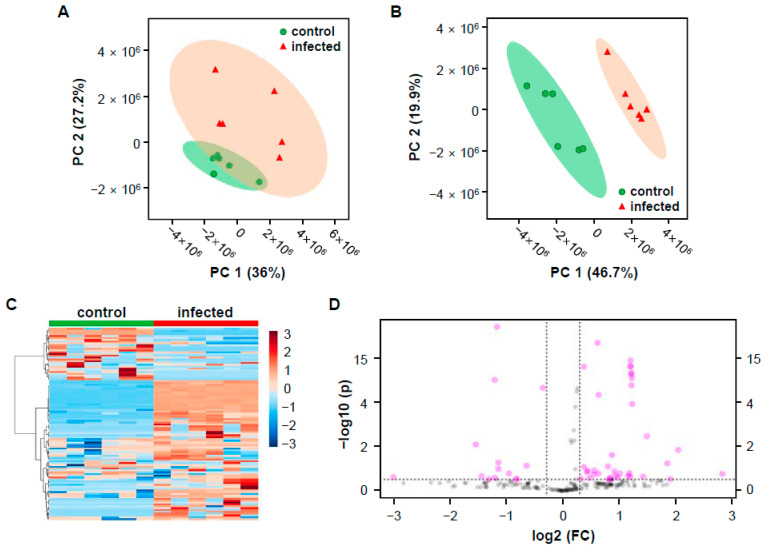
Multivariate statistical analysis of control and infected rice groups based on the mass features that generated by MZmine 2.53. (**A**) Principal component analysis (PCA) score plot with 95% of confidence regions. (**B**) Partial least squares discriminant analysis (PLS-DA) score plot with 95% of confidence regions. (**C**) Heatmap clustering result using Ward’s hierarchical clustering algorithm. (**D**) Volcano plot analysis. Important features were selected by volcano plot with fold change (FC) threshold (X axis) 2 and t-tests threshold (Y axis) 0.1. The red circles represent features above the threshold.

**Figure 3 plants-10-00213-f003:**
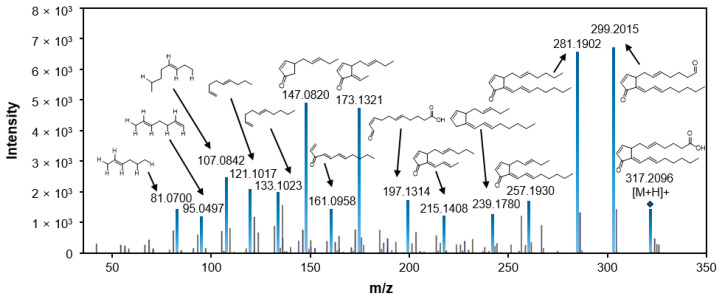
LC-MS/MS fragmentation pattern of 15-deoxy-Δ^12,14^-prostaglandin J2. For the identification, chemical formula and exact m/z value were generated by MassHunter software. MS/MS spectra were compared to mass spectra in databases such as Metlin, MassBank, HMDB, and GNPS.

**Figure 4 plants-10-00213-f004:**
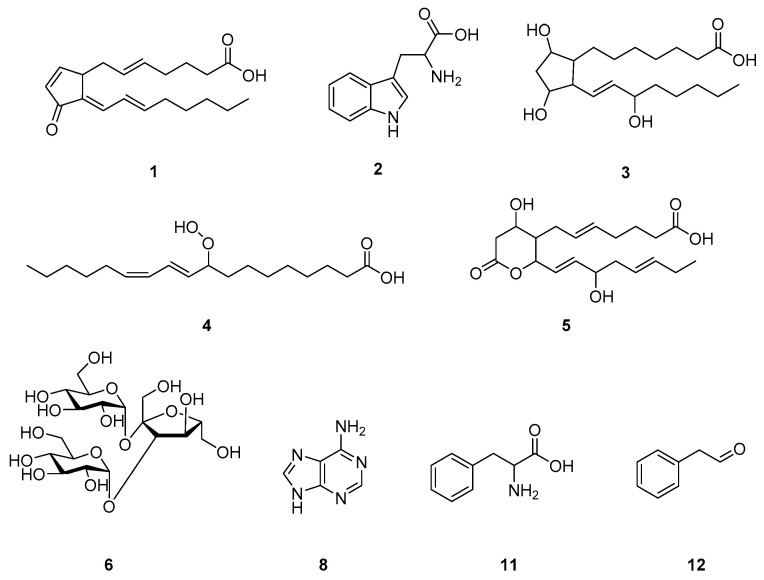
Potential chemical structures of nine metabolites most differentially induced in infected rice. (**1**) 15-deoxy-Δ^12,14^-prostaglandin J2; (**2**) tryptophan; (**3**) prostaglandin F1α; (**4**) 9-hydroperoxy-10*E*,12*Z*-octadecadienoicacid; (**5**) 11-dehydro-thromboxane B3; (**6**) melezitose; (**8**) adenine; (**11**) phenylalanine; (**12**) phenylacetaldehyde.

**Figure 5 plants-10-00213-f005:**
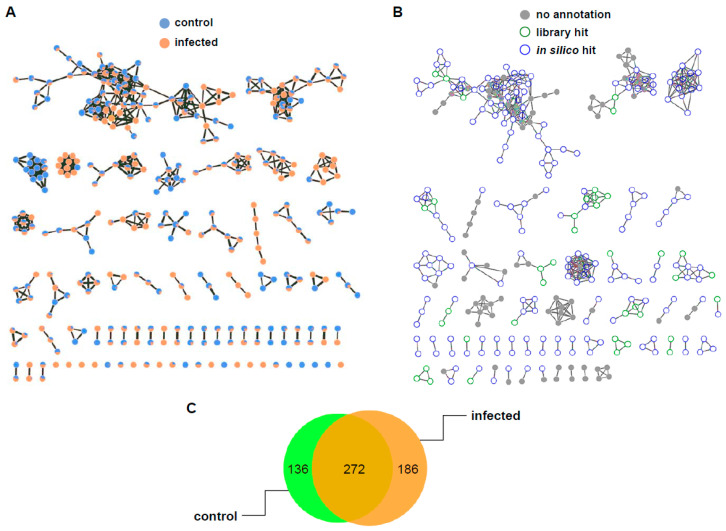
The networking analysis results of control and infected rice. (**A**) Visualized molecular networking. Network was generated using MS/MS spectra through classical molecular networking of GNPS server and visualized with nodes and edges through Cytoscape 3.8.0. Nodes consisted of pie charts based on peak intensity proportion in each metabolite. Blue, control; red, infected rice. The thickness of the edges was determined by the similarity of the two connected nodes with edge widths ranging from 7.5 to 15.0. (**B**) The network of the nodes annotated by network annotation propagation (NAP). Blue hollow nodes are represented by NAP consensus top ranked candidates annotated in silico. (**C**) Venn diagram of shared nodes for whole nodes in the classical molecular networking.

**Figure 6 plants-10-00213-f006:**
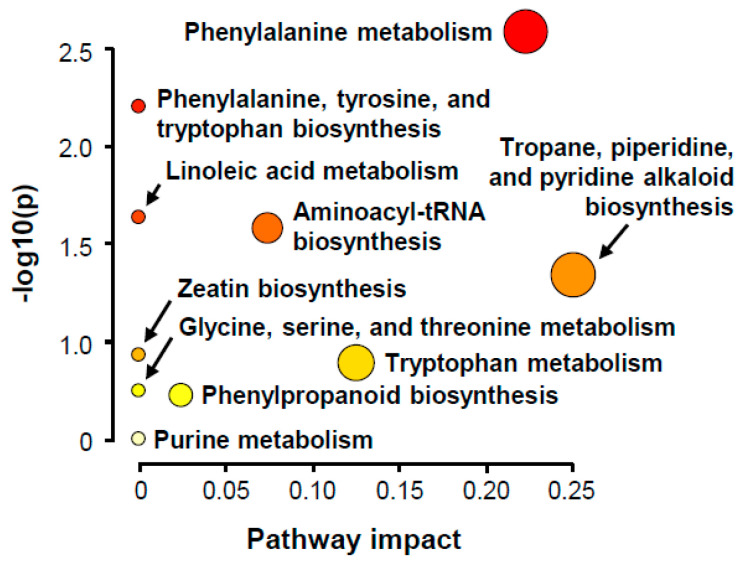
Summary plot of KEGG pathway analysis. Pathway analysis was conducted to determine biological processes and metabolic pathways of induced metabolites in infected rice. Out degree centrality was selected as node importance measure method for topological analysis. The color and size of each circle corresponds to its *p*-value and pathway impact, respectively.

**Table 1 plants-10-00213-t001:** List of metabolites most differentially induced in pathogen-infected rice compared with control.

No	*m*/*z*	RT ^a^ (min)	log2(FC)	−log10(*p*)	Representative MS/MS Fragments	Compound ID	Class	Adduct Ions
1	317.2096	13.64	20.503	15.026	107.0842 121.1017 147.0820 173.1321 299.2015	15-deoxy-Δ^12,14^-prostaglandin J2	prostaglandins	[M + Na]^+^
2	188.0696	3.39	19.939	14.257	118.0642 143.0719	tryptophan	amino acids	[M + H−NH_3_]^+^
3	357.2580	15.43	19.457	14.129	102.0867 176.9264 216.9466 256.9701 340.3596	prostaglandin F1α	prostaglandins	[M + H]^+^
4	313.2325	15.51	19.376	13.676	154.9296 196.9310 214.9464 256.9654	9-hydroperoxy-10*Z*,12*E*-octadecadienoic acid	octadecanoids	[M + H]^+^
5	330.2976	13.46	18.772	1.784	107.0844 121.1000 130.0855 264.2675 282.2745	11-dehydro-thromboxane B3	thromboxanes	[M + H−2H_2_O]^+^
6	527.1528	1.40	17.835	1.955	185.0411 203.0513 305.0828 365.1016	melezitose	trisaccharides	[M + Na]^+^
7	383.2025	16.89	16.834	1.673	104.9929 141.0741 156.0904 207.0625 267.1209	Unknown 1	oxadiazoles	-
8	136.0610	1.40	2.366	1.408	119.0338 109.0500	adenine	nucleobases	[M + H]^+^
9	303.1402	4.73	2.210	3.713	117.0324 127.0968 145.0266 177.0533	Unknown 2	-	-
10	699.3502	12.67	1.660	1.124	185.0405 294.9074 347.0933 537.2977 699.3497	Unknown 3	-	-
11	166.0854	2.37	1.252	10.862	103.0533 120.0795	phenylalanine	amino acids	[M + H]^+^
12	120.0802	2.37	1.219	16.785	103.0536	phenylacetaldehyde	phenylpropanoids	[M + H]^+^

^a^ RT, retention time.

**Table 2 plants-10-00213-t002:** Result from KEGG pathway analysis activated in pathogen-infected rice.

KEGG Pathway	Total ^a^	Hits ^b^	*p*-Value
Phenylalanine metabolism	12	2	1.83 × 10^−3^
Phenylalanine, tyrosine, and tryptophan biosynthesis	22	2	6.21 × 10^−3^
Aminoacyl-tRNA biosynthesis	46	2	2.60 × 10^−2^
Linoleic acid metabolism	4	1	2.26 × 10^−2^
Tropane, piperidine, and pyridine alkaloid biosynthesis	8	1	4.48 × 10^−2^
Zeatin biosynthesis	21	1	1.14 × 10^−1^
Tryptophan metabolism	23	1	1.24 × 10^−1^
Glycine, serine, and threonine metabolism	33	1	1.74 × 10^−1^
Phenylpropanoid biosynthesis	35	1	1.83 × 10^−1^
Purine metabolism	63	1	3.08 × 10^−1^

^a^ Total, total number of compounds in the pathway; ^b^ hits, actually matched number from the user uploaded data.

**Table 3 plants-10-00213-t003:** List of pathway modules containing ten detected KEGG pathways.

Pathway Module	KEGG Pathway
Amino acid metabolism	Phenylalanine metabolism
	Phenylalanine, tyrosine, and tryptophan biosynthesis
	Tryptophan metabolism
	Glycine, serine, and threonine metabolism
Biosynthesis of other secondary metabolites	Tropane, piperidine, and pyridine alkaloid biosynthesis
	Phenylpropanoid biosynthesis
Lipid metabolism	Linoleic acid metabolism
Biosynthesis of terpenoids and polyketides	Zeatin biosynthesis
Nucleotide metabolism	Purine metabolism
Genetic transcription	Aminoacyl-tRNA biosynthesis

## Data Availability

The data presented in this study are available on request from the corresponding author.

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
