# Peer review of "Application of UPLC-QTOF-MS Based Untargeted Metabolomics in Identification of Metabolites Induced in Pathogen-Infected Rice"

_plants, 2021, doi:10.3390/plants10020213_

Round 1

Reviewer 1 Report

In the manuscript authored by Mira Oha et al, the authors described a study based on an untargeted metabolomics approach to identify metabolites majorly involved in pathogen-infected rice compared to untreated rice.

Results are interesting and manuscript is well conducting. I appreciate the idea of using a machine learning-based multimodal classification approach as PCA and clustering analysis. Despite, I suggest authors to better describe in method section the clustering analysis represented with heatmap analysis. I suggest author in future to consider also decision trees and random forest as possible option to further dissect metabolites shared between infected and not infected rice.

Therefore, I suggest the manuscript for a possible publication after this minor correction and English editing revision will be performed.

Author Response

Answer: As the reviewer suggested, we described more detail methods about heatmap clustering analysis. And thank you for suggesting the application of decision trees and random forest to the dissection of metabolite shifts. We will apply them to another study.

Reviewer 2 Report

Dear editor/authors, the manuscript investigates the metabolomic profile of rice upon infection by Magnaporthe oryzae, by using a liquid chromatography quadrupole time-of-flight mass spectrometry system. In phytopathological terms of view, the manuscript presents the findings of just one experiment, without any in depth analysis of the identified metabolites and their meaning in plant defense. Therefore the results are limited and do not support a research study. It seems that the authors have focused on the methodology for obtaining data rather than trying to answer a scientific question. This is also obvious from the title of the manuscript that emphasizes on the technique rather than a phytopathological question. Therefore, I cannot recommend the acceptance of the manuscript due to the limited data and the absence of a phytopathological background and scope.

Author Response

Answer: Investigation on metabolic profile is necessary for in depth study of plant pathology or plant defense. Rice blast is a very common but highly important disease of rice, so various studies have been conducted. Even though few studies on the metabolic profile have been performed, it was difficult to see the changes in the overall metabolic profile with the conventional approach of separating and identifying the changed compounds. However, in our experiment, which applied ULPC-MS/MS platform and machine learning-based multimodal classification approach to verify and identify the metabolites, have allowed us to identify what changes have occurred in the overall metabolic profile of infected rice. Our research is more focused on metabolic changes in infected plants than on plant pathology. Therefore, our experiment is meaningful in the way by presenting new methodology that can be further used in phytopathology or plant defense study.

Reviewer 3 Report

I find the reviewed paper very interesting and suitable for publication in Plants.

I recommend some improvements (see below), before its acception.

Abstract

Delete “the large-scale study of small molecules”.

Rephrase to “… (HR-MS) is one of the most widely accepted metabolomics tools

Rephrase to “the role of these compounds in metabolic pathways was finally investigated using pathway analysis”

Rephrase “these results exhibited that” to “our study showed that…”

Replace “secondary metabolites” by “specialized metabolites”

Introduction

L52: Delete “primary and secondary metabolites” or replaced them by “specialized and non-specialized metabolites”

L53: Rephrase to “…provides understanding the plant…”

L67 Replace “,” by “;”

Results and discussion

L114: Replace “primary” by “non-specialized”

L116, L121: Replace “secondary” by “specialized” and all over the manuscript

L125: Replace by “ionized”

L186-188: Not understandable what the authors meant by that paragraph sentence.

L191& L194: Please replace “identified” to “tentatively identified”

What is the difference between Fig. 1 (A,B) and Fig S1? Please explain what is the new information, if any, we can get out of Fig. 1 vs Fig.S1?

Figure 4. Adjust its title from “representative chemical structures” to “potential chemical structures”

Table 1: Please include the “representative MS/MS fragments” from Table S1 into Table 1. Therefore, delete Table S1.

Materials and methods

L353-406: Please add a space between temperature values and its unit. (°C).

L387: I would like to understand what makes the analytical method “UPLC”? Based on the column and conditions, it looks to me like a regular HPLC method. What was the system backpressure during the run? It is true that Agilent 1290 can work as an UPLC pump, but I cannot see this from the operating conditions. If the system and analytical methods were used as a classical HPLC, please replace this all over the text, including the title.  

Author Response

Abstract

Delete “the large-scale study of small molecules”.

Rephrase to “… (HR-MS) is one of the most widely accepted metabolomics tools

Rephrase to “the role of these compounds in metabolic pathways was finally investigated using pathway analysis”

Rephrase “these results exhibited that” to “our study showed that…”

Replace “secondary metabolites” by “specialized metabolites”

Answer: As the reviewer commented, we rephrased the mentioned sentences and words.

Introduction

L52: Delete “primary and secondary metabolites” or replaced them by “specialized and non-specialized metabolites”

L53: Rephrase to “…provides understanding the plant…”

L67 Replace “,” by “;”

Answer: As the reviewer commented, we rephrased the mentioned sentences and words. 

Results and discussion

L114: Replace “primary” by “non-specialized”

L116, L121: Replace “secondary” by “specialized” and all over the manuscript

L125: Replace by “ionized”

L186-188: Not understandable what the authors meant by that paragraph sentence.

L191& L194: Please replace “identified” to “tentatively identified”

Answer: As the reviewer commented, we rephrased the mentioned sentences and words. However, in the case of the terms ‘primary and secondary’, which reviewers have mentioned to change to ‘non-specialized and specialized’, it seems more familiar to readers to write them as ‘primary and secondary’. Primary metabolites are the molecules that populate the pathways essential for life, secondary metabolites instead populate pathways that may only occur in some cells or in some organisms in some circumstances, for example when plants respond to pathogens by synthesis of defensive small molecules (Natural Product Biosynthesis, Walsh and Tang, 2017). Since what we wanted was to see the metabolites produced by the affected biosynthetic pathway in rice in certain circumstance, it would be more appropriate to write it as a secondary metabolite.

What is the difference between Fig. 1 (A,B) and Fig S1? Please explain what is the new information, if any, we can get out of Fig. 1 vs Fig.S1?

Answer: Although both Fig. 1 (A,B) and Fig. S1 are LC-MS chromatograms of same samples, they were performed in different ionization modes. The TIC of Fig. 1 (A,B) is ESI positive mode and Fig .S1 is ESI negative mode. Plant metabolites are differently ionized in each positive or negative ion mode, so the choice of which ESI mode will affect further results. As described in results section, our samples were more strongly ionized in the ESI positive mode than negative mode and we used the positive ion data in further analysis. Thus, we additionally showed Fig. S1 in supplementary data to show the evidence of using the positive mode in this study.

Figure 4. Adjust its title from “representative chemical structures” to “potential chemical structures”

Answer: As the reviewer commented, we changed the word.

Table 1: Please include the “representative MS/MS fragments” from Table S1 into Table 1. Therefore, delete Table S1.

Answer: As the reviewer commented, we changed the Table 1 and deleted Table S1.

Materials and methods

L353-406: Please add a space between temperature values and its unit. (°C).

Answer: As the reviewer commented, we added a space.

L387: I would like to understand what makes the analytical method “UPLC”? Based on the column and conditions, it looks to me like a regular HPLC method. What was the system backpressure during the run? It is true that Agilent 1290 can work as an UPLC pump, but I cannot see this from the operating conditions. If the system and analytical methods were used as a classical HPLC, please replace this all over the text, including the title.

Answer: As described in method section, we used YMC-Triart C18 column (2.0 × 150 mm, 1.9 μm) column, which developed for UHPLC analysis. Usually, the column of particle size 5 μm is used for the HPLC analysis. In this study, we used the column of particle size 1.9 μm with 100MPa internal pressure, which enables faster and higher separation. The system backpressure during the run was 700-800 bar.

Reviewer 4 Report

The manuscript is an interesting example of the application of ULPC-MS/MS platform to verify the possibility to identify the metabolites induced by the rice pathogenic infection.

The manuscript is well-structured and the authors use the tools actually available to elaborate the chemical features obtained through the LC-MS analyses. The manuscript deserves publication in Plants but I have some comments related to some of the putative induced metabolites and reported in table 1 as for example compound 10. The chemical class of this compound is not correct in table 1 because its name corresponds to a carbohydrate and not to a monoterpenoid. Are the authors sure about the correct identification of this compound? Do they have data from literature about its molecular weight and mass fragmentation data? Can they report them? Is it justifiable a so high retention time? Moreover, no explanation regarding its biosynthetic pathway and origin is reported for this compound unlike the other components whose origin has been explained. The same is for melezitose: it is included in the class of disaccharides while it is a trisaccharide. Are the authors sure about the correct identification? I suggest the authors to confirm the identification of the components reported in table with the co-injection of the commercial standards, when available, and for many of them the standard is available at low cost.

Another small comment is related to the use of the term adduct in Table 1: it is not very precise; it should be better to use the term: precursor ions (i.e. pseudomolecular or adduct ions)

Round 2

Reviewer 2 Report

I found satisfactory the reply of the authors to my comments.

Reviewer 4 Report

The revised version of the manuscript is now acceptable for publication in Plants